# Systematic Comparison of Extract Clean-Up with Currently Used Sorbents for Dispersive Solid-Phase Extraction

**DOI:** 10.3390/molecules29194656

**Published:** 2024-09-30

**Authors:** Michelle Peter, Christoph Müller

**Affiliations:** Department of Pharmacy, Center for Drug Research, Ludwig-Maximilians-Universität München, 81377 Munich, Germany; michelle.peter@cup.uni-muenchen.de

**Keywords:** dSPE, GC-MS/MS, matrix effects, pesticides, pollutants, QuEChERS

## Abstract

Dispersive solid-phase extraction (dSPE) is a crucial step for multiresidue analysis used to remove matrix components from extracts. This purification prevents contamination of instrumental equipment and improves method selectivity, sensitivity, and reproducibility. Therefore, a clean-up step is recommended, but an over-purified extract can lead to analyte loss due to adsorption to the sorbent. This study provides a systematic comparison of the advantages and disadvantages of the well-established dSPE sorbents PSA, GCB, and C18 and the novel dSPE sorbents chitin, chitosan, multi-walled carbon nanotube (MWCNT), and Z-Sep^®^ (zirconium-based sorbent). They were tested regarding their clean-up capacity by visual inspection, UV, and GC-MS measurements. The recovery rates of 98 analytes, including pesticides, active pharmaceutical ingredients, and emerging environmental pollutants with a broad range of physicochemical properties, were determined by GC-MS/MS. Experiments were performed with five different matrices, commonly used in food analysis (spinach, orange, avocado, salmon, and bovine liver). Overall, Z-Sep^®^ was the best sorbent regarding clean-up capacity, reducing matrix components to the greatest extent with a median of 50% in UV and GC-MS measurements, while MWCNTs had the largest impact on analyte recovery, with 14 analytes showing recoveries below 70%. PSA showed the best performance overall.

## 1. Introduction

In 2003, Anastassiades et al. [1] introduced a new sample preparation method for the analysis of pesticides in fruits and vegetables, the QuEChERS sample preparation method (acronym for “quick, easy, cheap, effective, rugged and safe”), which was better adapted to the modern requirements in pesticide analysis than previously common, often over 30-year-old methods [1]. Since then, the QuEChERS concept has been adopted in the EN 15662 [2] and in the AOAC’s official method 2007.01 [3] due to its ease of use and broad application. The triumph of this approach has continued for the analysis of environmental contaminants in food and feed matrices such as cabbage and radish [4], wastewater and sludge samples [5], and samples for contamination monitoring in the environment, of animal (e.g., bats [6,7], fish [8], or insects [9]) and plant origin (e.g., beech leaves and spruce needles [10], vine leaves [11], and forage grass [12]), or soil [13].

The QuEChERS method consists of two steps, a salt-assisted liquid–liquid extraction (SALLE), followed by a clean-up step through dispersive solid-phase extraction (dSPE). The use of SALLE is important to achieve rapid phase separation, to salt out proteins, and to maximize the extraction capacity for organic compounds (e.g., pesticides and organic pollutants) [1]. The subsequent clean-up step is necessary to remove a large part of co-extracted organic matrix components, in order to reduce matrix effects [14] and to optimize selectivity and sensitivity [15]. Ideally, this sample preparation grants a high reproducibility in combination with low analyte losses during the procedure. Contrary to previously applied methods, the QuEChERS concept implements dSPE for extract purification. The sorbent is dispersed in the extract during sample clean-up, in contrast to traditional solid-phase extraction (SPE), where the sorbent is packed in plastic cartridges. The original QuEChERS method uses ethylenediamine-*N*-propyl (PSA)-functionalized silica as a sorbent (Figure 1) [1]. PSA has already been used for sample preparation as a sorbent in SPE. Due to its weak anion exchange capacity, it retains polar organic compounds like acids and sugars [16], as well as metal ions through chelation [17].

With the alterations in the original QuEChERS method, different sorbents were used for dSPE. The EN 15662 [2] recommends the application of octadecyl-functionalized silica (C18; Figure 1) as well as graphitized carbon black (GCB; Figure 1) as alternative sorbents, depending on the matrix to be analyzed. For fatty matrices such as avocados or olives, the EN:15662 [2] recommends the addition of C18 to dSPE, to remove lipophilic matrix components (e.g., fatty acids). GCB is commonly applied to reduce the amount of large, planar molecules, like pigments (e.g., chlorophyll, carotenoids) and steroids in the matrix extract [18]. The combination of these three sorbents, PSA, C18, and GCB, has proven useful for several complex matrices, providing a solid clean-up with good reproducibility and high recovery rates [6,19,20,21]. However, the application of GCB for matrix clean-up can negatively influence the recovery of larger, planar analytes [22,23], namely certain pesticides (e.g., thiabendazole; Appendix A) but also steroid hormones (e.g., testosterone or estradiol; Appendix A) or polycyclic aromatic hydrocarbons (PAHs; Appendix A).

Apart from these three sorbents, recent QuEChERS-based methods applied different sorbents that were not part of the regulatory method. Multi-walled carbon nanotubes (MWCNTs) are coaxial closed layers of carbon hexagons with two to fifty nested into one another (Figure 1). They were originally introduced by Sumio Iijiama in 1991 [24] and are mainly used in semiconductor technology. Due to their large surface area and unique absorption properties, MWCNTs are described as a possible alternative to GCB. Both are able to establish π–π interactions and therefore remove nonpolar compounds during sample clean-up [25]. MWCNTs are used for the analysis of pesticides in garlic [26], fruit juices [27], as well as fruit and vegetables [28].

Z-Sep^®^ is a quite recently introduced, commercially available sorbent based on zirconium dioxide-functionalized silica (Figure 1) [29]. According to the supplier, it is suitable for the sample clean-up of fatty matrices when highly lipophilic analytes are of interest, through the removal of lipids and pigments via Lewis acid–base interactions [30]. It has been successfully applied to multiresidue analysis in fish [31], extending the scope of the QuEChERS concept to flame retardants and other emerging environmental pollutants like PAHs.

With growing interest in green chemistry, the applications of biopolymers as sorbents in sample clean-up became popular, as they are available from renewable sources with high prevalence. The glucose derivative chitin, a polymer of *N*-acetylglucosamine (Figure 1), is the second most abundant biopolymer in nature and therefore easily available [32]. It can be isolated through processing the discard of crustaceans like crawfish [33], adding value to crustacean shell waste, since shellfish discard and its deposal pose a challenge for shellfish-producing countries [34]. The use of chitin as a sorbent in QuEChERS-based analytical methods could increase recoveries for pesticides, active pharmaceutical ingredients (APIs), and personal care products (PCPs) in drinking water treatment sludge [35]. Chitin and its deacetylated form chitosan (Figure 1) [34] have been successfully used to adsorb metal ions for the detoxification of water [36]. Due to their molecular structures and functional groups (hydroxyl, amine, or amide), chitin and chitosan are effective for the removal of several organic matrix components, including dyes [37] and lipids [38], similar to PSA.

Moreover, there are many more different sorbents described for their application in sample preparation. Recent developments for solid-phase extraction (SPE) include mesoporous silica which often needs the addition of surface modifiers [39], or sorbents based on magnetic nanoparticles (MNPs) and molecular imprinted polymers (MIPs) [40]. The advancements in sorbent development and availability underline the strong interest in dSPE as a sample clean-up and increase its meaning and scope [40].

However, the different sorbents only get tested for different matrices when they are individually of interest in certain matrices. In this study, we systematically compared seven of the most prominent sorbents in dSPE, PSA, C18, GCB, chitin, chitosan, MWCNTs, and Z-Sep^®^, regarding their clean-up capacity. We also assessed the recoveries they achieved for a broad range of 98 analytes, including 51 pesticides, 12 active pharmaceutical ingredients (APIs), 12 persistent organic pollutants (POPs), four polycyclic aromatic hydrocarbons (PAHs), 11 personal care product ingredients (PCPs), three flame retardants, two plasticizers, two industrial chemicals, and one indicator for anthropogenic pollution in five different common matrices (spinach, avocado, orange, bovine liver, salmon), to achieve a better understanding of the ideal scope and limitations of each sorbent. For all the chosen matrices, adapted and validated QuEChERS-based analytical methods for pesticide and POP quantification were published [6,22,41,42,43,44,45]. The matrices were chosen to preferably cover a broad scope of applications of the QuEChERS approach. They are of plant (avocado, spinach, and orange) and animal (bovine liver, salmon) origin and cover lipid-dense as well as high-water, high-protein, and/or high-pigment content matrices (Table 1). The different compositions of the chosen matrices pose a variety of challenges to sample clean-up. Spinach extract is especially rich in coloring matrix components, mainly chlorophylls and certain carotenoids and flavonoids, requiring a refined clean-up [41]. Contrary to most plant-based matrices, avocado has a relatively high lipid content (15%), thus posing different challenges to sample preparation [46]. In the avocado, as well as in salmon and liver, the high fat content contributes to the complexity of the matrix, being easily co-extracted during SALLE, thus negatively affecting method performance unless removed from the extract with dSPE [47]. The matrix components in citrus fruits like the orange consist of a variety of flavonoids, coumarins, polyphenols, as well as terpenoids [48]. Furthermore, the citric acid in orange extracts leads to a slightly acidic pH [49], which influences the recovery rate especially of basic analytes. The avocado as well as the orange are part of the most studied fruits in the last years regarding pesticide burden [50]. Liver and fish muscle are especially interesting matrices because lipophilic contaminants (like POPs) can accumulate in the fatty tissue [51].

The tested analytes also covered a broad range of analytes commonly assessed with QuEChERS methods. They included pesticides, which the QuEChERS approach was originally intended for [1], as well as substance classes that were since included in QuEChERS-based methods, such as PAHs [52], PCPs [53] and APIs [54]. These substances are of particular concern in environmental and food analysis. Pesticides and POPs (such as lindane, polychlorinated biphenyls, DDT) are prone to bioaccumulation along the food chain and have been shown to negatively affect the environment, for example by decreasing reproduction, showing endocrine-disruptive effects, and overall increasing mortality rates when found in mammals [55,56]. PAHs, like fluorene or phenanthrene, have been of interest due to their carcinogenic and mutagenic potential [57]. Substances like the UV blocker avobenzone were assessed by the European Chemicals Agency (ECHA) under their community rolling action plan (CoRAP) for reasons like high aggregated tonnage, high exposure, and potential persistency, bioaccumulation, and toxicity (PBT) [58]. APIs, including the opioid codeine [59], the anticonvulsant carbamazepine [60], or the hormone medication 17α-ethinylestradiol [61], have been found in wastewater and drinking water samples. Exposure to APIs can have negative impacts on the environment even in low concentrations, due to the unknown synergistic effects and long-term exposure [59,60,61]. The UV filter oxybenzone was shown to have genotoxic effects on corals and is a known endocrine disruptor in mammals and fish [62].

With the combination of matrices, analytes, and sorbents tested in this study, we aimed to cover a variety of possible applications of the QuEChERS sample preparation approach and provide a profound overview of which commonly used sorbent would be suitable for different scopes of applications.

**Table 1 molecules-29-04656-t001:** Distributions of water contents, total lipids (fats), carbohydrates, proteins, and other organic components in the analyzed matrices.

Matrix	Nutrients	Rich in
spinach [63] (mature)	92% water 3% carbohydrates 3% proteins	chlorophyll vitamin A, C, and K carotenoids and flavonoids
avocado [64] (raw)	73% water 15% total lipids (fats) 9% carbohydrates	phytosterols monounsaturated fatty acids carotenoids
orange [65] (raw)	87% water 12% carbohydrates	vitamin C carotenoids citric acid
salmon [66] (Atlantic farm-raised, raw)	66% water 20% proteins 13% total lipids (fats)	polyunsaturated fatty acids vitamin B12 and D cholesterol
liver [67] (pork)	71% water 21% proteins 4% total lipids (fats)	cholesterol bile acids vitamin A, B12, and D

## 2. Results and Discussion

### 2.1. Determination of the Sample Clean-Up Capacity

#### 2.1.1. By Visual Inspection

In the first step, the raw and cleaned-up extracts were visually examined. Figure 2 shows the colors of each extract after clean-up with different dSPE sorbents. The colors were taken from the pictures shown in Appendix A, for better comparison. The spinach extracts showed no difference in color or intensity except after clean-up with PSA, where they had a lighter, less intense green color. For the orange, avocado, salmon, and liver, the sample clean-up with both GCB and MWCNTs led to a nearly colorless extract, significantly reducing the coloring matrix components, as they are commonly applied to remove large nonpolar molecules like pigments. In the avocado extracts, the application of PSA and Z-Sep^®^ also slightly reduced the intensity of the color of the extract. This effect was also observed for liver extracts, where PSA and Z-Sep^®^, and additionally chitosan, removed most coloring matrix components. For the salmon extracts, a reduced intensity was also present after the clean-up with Z-Sep^®^. The colors were extracted from the pictures shown in Appendix A, where the blank value represents the color of the background. The color transition between Figure 2 and Appendix A was achieved by extracting the color of the respective matrix extract from Appendix A via the Microsoft Office PowerPoint 365’ eyedropper tool. The colors in Figure 2 have the same RGB code as the according matrix extracts in Appendix A.

#### 2.1.2. By UV Measurements

Besides visual inspection, the ability of the seven dSPE sorbents to effectively clean-up raw matrix extracts was determined by UV measurements. As many matrix components absorb at 220 nm, a lower relative extinction corresponded to a cleaner extract, where components absorbing at the specific wavelength were removed. Figure 3 shows the relative extinction of extracts from spinach, avocado, orange, salmon, and liver, each cleaned-up with PSA, C18, Z-Sep^®^, chitin, chitosan, GCB, or MWCNTs. The distribution of relative extinctions across all matrices (Figure 3A) showed the lowest relative extinction for extracts cleaned-up with PSA and Z-Sep^®^. This correlated quite well with the observations from the visual inspection, indicating that both PSA and Z-Sep^®^ could remove visible and UV active components from a variety of matrices. However, despite the visible color reduction, the relative extinction after clean-up with GCB and MWCNTs was quite high. The same applied to C18, whereas chitin and chitosan led to moderate relative extinctions. Regarding the detailed distribution of relative extinctions (Figure 3B), it became clear that spinach had the lowest relative extinctions across all sorbents except for Z-Sep^®^, where avocado yielded the lowest value. This showed that the sorbents were able to remove plenty of matrix components form the spinach extracts, despite nearly no changes in the color intensity of the extracts. The best performing sorbents, regarding the removal of matrix components that absorbed at 220 nm, were PSA for spinach and liver, and Z-Sep^®^ for the avocado, orange, and salmon. C18 removed neither the coloring agents from the extracts nor the matrix components adsorbing at 220 nm. GCB and MWCNTs were not able to remove matrix components absorbing at 220 nm, despite removing most coloring agents from the extracts. This underlines the fact that visual colorless extracts are not necessarily clean in terms of containing interfering matrix components.

#### 2.1.3. By MS Measurements

As an additional method to assess the clean-up capacity of the sorbents, the total ion current (TIC) chromatograms of blank and cleaned-up extracts were compared. Appendix A shows the TIC chromatograms of each matrix cleaned-up with each sorbent in comparison to the corresponding raw matrix. The peak areas of each matrix cleaned-up with each sorbent are summarized in Appendix A. In Figure 4, the relative TIC peak areas of cleaned-up extracts compared with raw extracts are shown. The distributions of relative peak areas across all matrices (Figure 4A) showed a similar pattern as in the UV measurements (Figure 3A), underlining that PSA and Z-Sep^®^ were able to remove a broad range of matrix components, including gas chromatography-mass spectrometry (GC-MS) detectable components. The other sorbents showed notably higher relative peak areas. The detailed comparison of relative peak areas (Figure 4B) showed that in general, all sorbents were able to remove approximately 50% or more of GC-MS detectable matrix components in the spinach and liver matrices, with C18 reducing the relative peak area for spinach and liver the least. The relative peak areas obtained from the cleaned-up orange extracts also showed quite considerable reduction compared to the raw extracts. Here, PSA, Z-Sep^®^, chitin, as well as chitosan performed well. The extracts from the salmon matrix only showed a clear reduction in the relative peak area after clean-up with Z-Sep^®^ and PSA. The avocado extracts generally had the lowest reduction in its relative peak areas, with clear effects only after clean-up with PSA, Z-Sep^®^, and C18. The best performing sorbent regarding the removal of matrix components detectable with GC-MS was Z-Sep^®^ for all matrices.

On the one hand, the results regarding the GC-MS detectable matrix components correlated quite well with the UV measurements, as in both experiments, PSA and Z-Sep^®^ reduced the relative signals (either peak area or extinction at 220 nm) to the largest extent. C18, chitin, and chitosan showed a moderate clean-up capacity, whereas especially GCB and MWCNTs were not able to reduce matrix components in considerable amounts. 

On the other hand, the sorbents did not remove UV-absorbing matrix components as well as they did GC-MS detectable components in the orange and salmon matrices. All sorbents, except PSA and Z-Sep^®^, reduced the relative peak area to a larger extent than they reduced the relative extinction at 220 nm, indicating that they removed more GC-MS assessable matrix components than UV-absorbing components. The relative peak areas obtained from the avocado extract could not be lowered through clean-up with chitin, chitosan, GCB, and MWCNTs.

### 2.2. Determination of the Analyte Recovery

Besides the removal of interfering matrix components, a crucial quality characteristic was the recovery of analytes. Ideally, the sample preparation would not remove analytes of interest while still providing a relatively clean extract. Guidelines for method validation like SANTE/11312 [68] or CXG 90-2017 (Guidelines on Performance Criteria for Methods of Analysis for the Determination of Pesticide Residues in Food and Feed) [69] define the parameters to evaluate this effect as recovery. For an analytical method to meet validation criteria, the recovery must lie within a range of 70 to 120% (SANTE/11312 [68] and CXG 90-2017 [69]). In this study, the influence of PSA, C18, Z-Sep^®^, chitin, chitosan, GCB, and MWCNTs on the recovery of the 98 analytes (Appendix A) with different physicochemical properties was tested to gain a better understanding of their ideal applications in sample preparation depending on the analytes of interest and sample matrix. Figure 5 shows the distribution of recoveries of all analytes across all five matrices obtained after sample clean-up with different dSPE sorbents. It is striking that while most sorbents achieved a median recovery well around 100% (from 94% for PSA to 104% for chitin), the application of MWCNTs led to clearly lower recoveries (median 86%) than the rest of the sorbents. However, when calculating the mean across all analytes, every sorbent for every matrix laid within the required range of 70 to 120% (Figure 5, upper left).

A more detailed analysis showed the influence of the different sorbents on the analytes spiked into the individual matrices (Figure 5). For all sorbents, the mean recovery as well as the box, which equaled the middle 50% of the data, laid within the recommended range of 70 to 120%. In this study, special interest was paid to analytes with recoveries below 70%, as the sorbents were especially accountable for these low values. A list of all analytes and their respective recoveries is provided in Appendix A.

After clean-up with chitosan, no analyte had a recovery below 70% and only very few analytes reached recoveries above 120%. Chitosan therefore did not negatively affect the analyte recoveries in the five tested matrices, so it is a suitable sorbent for a broad range of applications.

Its structural derivative, chitin, showed a similar performance only with one analyte (codeine, 68%; Appendix A) below 70% recovery in the spinach extract. However, as a recovery of 68% was only slightly under the criterium, chitin could nonetheless be a suitable sorbent for the analysis of codeine and structurally familiar components.

Another sorbent that led to mostly recommended recoveries with a low scatter around the median was PSA. Only in the spinach, salmon and liver matrices did some analytes show recoveries below 70% (clotrimazole 66% in spinach, lilial 52% in spinach, 62% in salmon, and 68% in liver, and avobenzone 67% in salmon (structures of the analytes are given in Appendix A)). As a weak anion exchange agent, PSA is commonly used to remove polar organic matrix compounds, such as acids and sugars [16]. Both lilial and avobenzone are susceptible to interactions with PSA. Lilial as an aldehyde could potentially form a covalent imine bond with the amine groups in PSA [70].

Avobenzone in general showed rather low recoveries after sample clean-up with PSA (from 75% in spinach to 90% in orange). The distributions of recoveries for C18 were quite similar to PSA; however, there were different analytes that did not comply with the recovery criteria. In the spinach extract cleaned-up with C18, the outlier was the herbicide isoproturon, with a recovery of 59%, and in the salmon extract, the herbicide atrazine had a recovery of 59%.

After clean-up with Z-Sep^®^, more analytes showed recoveries below 70%, and the outliers were further away from the recommended recovery. Avobenzone had recoveries from 12% in salmon extract to 48% in spinach extract, and only in the orange extract was 73% reached. Through its tautomerism, the 1,3 diketone avobenzone is an ideal bidentate ligand for the zirconium in Z-Sep^®^, which adsorbs avobenzone to a greater extent than PSA. The other outliers included benzophenone-1 (68% in salmon, 69% in avocado), codeine (65% in spinach, 66% in orange, 68% in salmon), imidacloprid (63% in avocado), and oxybenzone (68% in salmon). Remarkably, besides avobenzone, the two ketones benzophenone-1 and oxybenzone had low recoveries, as the oxygen of the ketol group (hydroxy ketone group) can act as a Lewis base and likely interacted with Z-Sep^®^, forming a Lewis acid–base adduct. Interestingly, PSA only affected the recovery of avobenzone, but not of other ketones (benzophenone-1 or oxybenzone), while Z-Sep^®^ had a lesser influence on the recovery of lilial. The interaction between ketones and Z-Sep^®^ however seems to have been weaker than between the β-diketone and Z-Sep^®^, as their recoveries were still close to 70%.

The sorbent GCB led to recoveries below 70% for avobenzone, phenanthrene, pyrene, and tetraconazole. Besides avobenzone, which seemed like a difficult analyte for most sorbents, the analytes affected by GCB were planar molecules with multiple aromatic rings that could easily form π–π interactions with the sp²-hybridized carbons of the graphene structure [71]. Pyrene had by far the lowest recoveries (from 19% in salmon to 58% in orange), and only in spinach did the recovery of 77% comply with the requirements. This might be due to the large amounts of pigments present in the spinach extract, potentially masking more active sites of the sorbent. This highlights the impact GCB can have on planar analytes during sample clean-up.

However, the lowest recoveries were clearly obtained when MWCNTs were used as sorbent for dSPE. A total of 14 analytes had recoveries below 70% in at least one of the matrices after clean-up with MWCNTs. Like with GCB, the large planar analytes showed the lowest recoveries, but the offset was significantly higher. Pyrene for example was not even detectable in the avocado, salmon, and liver extracts after clean-up with MWCNTs, and it had recoveries of 4% in the orange and 10% in the spinach extracts. Other planar analytes, such as fluorene, phenanthrene, and thiabendazole, were severely affected as well. Furthermore, avobenzone, benzophenone-1, and oxybenzone also had low recoveries, but since they were also affected by other sorbents, this might be due to other factors of the clean-up procedure, possibly the slightly acidic pH of the citrate-buffered extraction system [3], which was not perfectly suitable for these analytes. Regarding the severe impact of MWCNTs on a large number of analytes in comparison to GCB, it became clear that GCB was the better option for the removal of large, planar matrix components. Both sorbents performed equally well in reducing the amounts of pigments from colored extracts and possibly other planar matrix components, but GCB did not impact the recovery of planar analytes of interest to the same extent as MWCNTs did.

## 3. Materials and Methods

### 3.1. Materials

#### 3.1.1. Chemicals

All analytes and the deuterated internal standards (azoxystrobin-*d*_4_, chlorpyrifos-*d*_4_, and pyrene-*d*_10_) had a purity >98% and were purchased either from BLDpharm (Kaiserslautern, Germany), ChemPUR (Karlsruhe, Germany), EDQM (Strasbourg, France), HPC Standards GmbH (Cunnersdorf, Germany), Merck (Darmstadt, Germany), or Tokyo Chemical Industries (TCI, Tokyo, Japan). A complete list of all 98 analytes is provided in Appendix A. Dispersive SPE bulk sorbents of ethylenediamine-*N*-propyl-functionalized silica (PSA), octadecyl-functionalized silica (C18), and graphitized carbon black (GCB) were obtained from Agilent Technologies (Santa Clara, CA, USA). The bulk of the multi-walled carbon nanotubes (MWCNTs, outside diameter 20–30 nm) was purchased from abcr (Karlsruhe, Germany). Chitin and chitosan (unknown purity) were obtained from TCI (Tokyo, Japan). Z-Sep^®^ was bought from Merck (Sant Louis, MO, USA). Acetonitrile (ACN) and isopropanol of HPLC grade were purchased from VWR (Darmstadt, Germany). 3-Ethoxy-1,2-propanediol (98%) was obtained from BLDpharm (Kaiserslautern, Germany), and L-gluconic acid *γ*-lactone (95%), D-sorbitol (99%), sodium citrate dihydrate (≥99%), and disodium hydrogen citrate sesquihydrate (≥99%) were purchased from Merck (Darmstadt, Germany). Shikimic acid (≥98%) was obtained from Carl Roth (Karlsruhe, Germany). Anhydrous magnesium sulphate (MgSO_4_, ≥98%) was purchased from Grüssing (Filsum, Germany). Sodium chloride (NaCl, p.a.) was obtained from Bernd Kraft (Duisburg, Germany).

#### 3.1.2. Matrices

For the preparation of blank matrices, the spinach, orange, avocado, salmon and bovine liver were bought at a local grocery store. Each matrix was homogenized with an Ultra Turrax^®^ T25 basic (IKA Labortechnik, Staufen, Germany) at 13,000 rpm for 15 min. After each matrix, the rotor was cleaned for 15 min with ultrapure water and isopropanol. After homogenization, the matrices were weighed into 50 mL centrifuge tubes in aliquots of 10.0 g (±0.5 g) (spinach and orange, according to EN:15662 [2]) or 5.0 g (±0.1 g) for the avocado [2], salmon, and liver [6], respectively. Where applicable, the amount of matrix per aliquot was corresponding to the EN:15662 [2]. For the liver, 5.0 g was used according to Schanzer et al. [6], and an equal amount was used for the salmon to ensure comparability. The homogenized matrix aliquots were stored in a freezer at –20 °C.

### 3.2. Reagents

#### 3.2.1. Stock Solutions

Stock solutions with a concentration of 1 mg mL^−1^ were prepared for each analyte and for the two internal standards azoxystrobin-*d_4_* and chlorpyrifos-*d_10_*. The stock solutions were combined into five working solutions, containing analytes grouped by their chemical properties (alcohols, acids, nonpolar substances, amines/amides, and ethers/esters), with a concentration of 10 µg mL^−1^ each [72]. An internal standard working solution was prepared accordingly. The stock and working solutions were stored at –20 °C and allowed to settle to room temperature for 1 h before use.

#### 3.2.2. Analyte Protectant Solution

The analyte protectant solution (AP) was prepared from 3-ethoxy-1,2-propanediol (200 mg mL^−1^), L-gluconic acid *γ*-lactone (10 mg mL^−1^), sorbitol (5 mg mL^−1^), and shikimic acid (5 mg mL^−1^) in ACN and water (6:4, (*v*/*v*)) according to an application note from the EU Reference Laboratories for Residues of Pesticides (EURL-SRM) from 2013 [73].

#### 3.2.3. Buffer Salt Mixture

The buffer salt mixture for the salting out step of the QuEChERS sample preparation consisted of anhydrous MgSO_4_, NaCl, sodium citrate dihydrate, and sodium hydrogen citrate sesquihydrate in a ratio of 8:2:2:1 according to EN:15662 [2]. Therefore, 4.00 g (±0.20 g) anhydrous MgSO_4_, 1.00 (±0.05 g) NaCl, 1.00 (±0.05 g) sodium citrate dihydrate, and 0.50 g (±0.03 g) disodium citrate dihydrate were weighed into 15 mL centrifuge tubes.

#### 3.2.4. dSPE Mixture

For the preparation of the dSPE mixtures, anhydrous MgSO_4_ and each respective sorbent were combined in 15 mL centrifuge tubes and vigorously shaken by hand for 1 min, except for the mixture containing GCB, which was mixed with anhydrous MgSO_4_ with a mortar and pestle. The ratio of anhydrous MgSO_4_ to sorbent was 3:1 (for PSA, C18, Z-Sep^®^, chitin, and chitosan) and 20:1 for GCB and MWCNTs [2]. Chitin and chitosan were washed/cleaned with *tert*-butyl methyl ether (4 times) and ACN (4 times) and evaporated to dryness with a rotary evaporator. Detailed information on the amounts of sorbent and anhydrous MgSO_4_ used per mL extract for clean-up is given in Table 2.

### 3.3. Instrumentation

#### 3.3.1. Sample Preparation Equipment

For sample homogenization, an Ultra Turrax^®^ T25 basic (IKA Labortechnik, Staufen, Germany) was used. For the centrifugation of the 50 mL centrifuge tubes, a Heraeus Megafuge (VWR, Darmstadt, Germany) was used. Centrifugation of the 15 mL centrifuge tubes was performed with an EBA 20 centrifuge (Hettich, Tuttlingen, Germany) and for 2 mL microcentrifuge tubes, a 5415 D centrifuge from Eppendorf (Hamburg, Germany) was used. Shaking steps were performed with a Vortex Genie (Scientific Industries, Bohemia, NY, USA).

#### 3.3.2. UV–Vis Spectrophotometer

The clean-up capacity of the sorbents was assessed with UV measurements at 220 nm, performed with an M2e plate reader from Molecular Devices (San Jose, CA, USA).

#### 3.3.3. Gas Chromatography-Tandem Mass Spectrometry (GC-MS/MS)

Analysis was performed with a 7890 gas chromatograph coupled with a 7010B triple quadrupole mass spectrometer, which was equipped with a high-efficiency ion source (HES) (all from Agilent, Santa Clara, CA, USA). For the injection of 1 µL, a PAL3 RSI autosampler from CTC analytics (Zwingen, Switzerland) was used. The columns for separation were two Agilent J&W HP 5MS Ultra Inert capillary columns (15 m × 250 µm × 0.25 µm) coupled with a capillary flow technology (CFT) backflush device. The GC was equipped with a multi-mode inlet (MMI), operated in solvent vent mode. All parameters of the GC-MS/MS method were applied as in our previously published QuEChERS approach [72]. A summary of the multiple reaction monitoring (MRM) conditions for all 98 analytes is listed in Appendix A.

### 3.4. Methods

All experiments were performed for all seven tested dSPE sorbents with the five matrices for spinach, avocado, orange, liver, and salmon, in triplicates.

#### 3.4.1. QuEChERS Sample Preparation

The sample preparation of the plant-based matrices of spinach, avocado, and orange, was performed according to the EN:15662 [2]. As the sample preparation of matrices of animal origin, like salmon and liver, is not specified in the EN:15662 [2], the clean-up step was performed according to the procedure for the analysis of pesticides and persistent organic pollutants (POPs) in animal liver tissues described by Schanzer et al. [6]. A defined portion (Table 3) of the homogenized matrix was weighed into a 50 mL centrifuge tube and an aliquot of ultrapure water was added (Table 3). Then, the samples were shaken with a Vortex Genie for 30 s, and 10 mL ACN were added before another shaking step of 30 s, followed by 15 min of extraction time. Afterwards, the buffer salt mixture was added, and the samples were instantly shaken by hand to prevent the formation of agglomerates. The samples were then again shaken with a Vortex Genie and centrifuged for 5 min at 3600× *g*. The organic upper layer (raw extract) was used for further clean-up with different dSPE sorbents (Table 2). Therefore, an aliquot of 1 mL extract was transferred to a 2 mL microcentrifuge tube containing 200 mg or 157.5 mg of dSPE mixture as described in Chapter 3.2.4. The sample was again shaken by hand followed by 30 s shaking with a Vortex Genie and then centrifuged for 5 min at 12,000× *g*. Aliquots of the supernatants were taken for further analysis. For the determination of analyte recovery, blank extracts (every matrix cleaned-up with every sorbent) were spiked one time before and one time after clean-up with 50 µL (before) or 5 µL (after) of a standard mix containing all analytes at 10 µg mL^−1^ to obtain final nominal concentrations of 50 ng mL^−1^. Extracts for other experiments were not spiked with analytes.

#### 3.4.2. Determination of the Sample Clean-Up Capacity

##### By Visual Inspection

After sample preparation and sample clean-up, the colors of the final extracts from each sorbent and matrix were visually examined.

##### By UV Measurements

For the assessment of the clean-up capacity of the different sorbents, the extinction at 220 nm wavelength of the cleaned-up extracts was compared to the raw extracts taken from the sample preparation before the dSPE step. Pure ACN was measured as a blank. Three hundred microliters of the solvent, raw extracts, and cleaned-up extracts were transferred to a 96-well fused silica plate and extinction at 220 nm was measured. The relative extinction was calculated according to Equation (1):(1)$$relative\;extinction\;[\%]=\frac{extinction\;({clean}_{220nm})}{extinction\;({raw}_{220nm})}\times 100$$

##### By MS Measurements

Additionally, extracts of each matrix cleaned-up with each sorbent were analyzed with GC-MS/MS used as described in Chapter 3.3.3. Instead of the MRM mode, the scan mode from *m/z* 100 to 600 was used and the total ion current (TIC) was summed up for every extract. The peak area was determined by using Agilent’s Agile2 integration algorithm in the Qualitative Analysis Navigator program (version B.08.00 from 2016).

#### 3.4.3. Determination of the Analyte Recovery

To determine the influence of different sorbents on analyte recovery, the extract of each matrix was spiked before and after the sample clean-up with each of the seven sorbents (PSA, C18, Z-Sep^®^, chitin, chitosan, GCB, and MWCNTs). The samples were measured with GC-MS/MS and recovery was calculated according to Equation (2):(2)$$recovery=\frac{area\;(after)}{area\;(before)}\times 100$$

## 4. Conclusions

The clean-up capacity of each sorbent (PSA, C18, Z-Sep^®^, chitin, chitosan, GCB, and MWCNTs) was assessed in three different ways: (1) by evaluating the color intensity of the cleaned-up extracts; (2) by measuring the relative extinction at 220 nm; and (3) by comparing the summed TIC peak areas. Considering all three parameters, Z-Sep^®^ showed the greatest clean-up capacity of the tested sorbents, reducing the GC-MS detectable matrix components in all five matrices to the largest extent. It also performed well in the UV experiment, removing most of the matrix components in the avocado, orange, and salmon extracts. This showed that Z-Sep^®^ was able to remove a broad range of different matrix components. Regarding analyte recovery, Z-Sep^®^ showed a negative influence on many analytes, second only to MWCNTs. The exact identification of the removed matrix components was not possible with the experimental setup. However, the aim of this study was to assess the capacities of different sorbents to reduce co-extracted matrix components while not negatively affecting analyte recovery. PSA provided the second-best result regarding clean-up capacity, while only negatively affecting the recovery of fewer, particular analytes, resulting in overall the best performance of all tested sorbents. The sorbents with a medium clean-up capacity (chitin and chitosan) especially had the least negative influence on the recovery rates of all tested analytes. GCB and MWCNTs only performed well in the visual inspection of color intensities, with contrary results in the other clean-up capacity experiments. Considering analyte recovery, GCB also showed mediocre results. Large planar molecules such as PAHs (see Appendix A) showed low recoveries when GCB or MWCNTs were used. In general, analytes with one or more (hydroxyl-) carbonyl groups, such as lilial or avobenzone (see Appendix A) were particularly susceptible to recovery loss. Overall, this study showed that the choice of the ideal sorbent is strongly dependent on the purpose of the applied method (Figure 6). If analytes that are difficult to access are of interest, the use of Z-Sep^®^ and GCB might not be sufficient, as they can lead to very low recoveries. Instead, chitin or chitosan could be viable options, providing a good compromise between clean extracts and few negative effects on analytes. If analyte recovery is not the crucial point, but clean-up is more important, Z-Sep^®^ is the best choice. PSA, as the most established dSPE sorbent, is still the best choice for most QuEChERS applications. The implementation of MWCNTs to QuEChERS-based sample preparation does not seem beneficial in any case. For complex matrices and difficult analytical purposes, individual consideration and a combination of different sorbents is necessary and cannot be replaced by this study. However, the published data provide a valuable starting point for future QuEChERS method development.

## Figures and Tables

**Figure 1 molecules-29-04656-f001:**
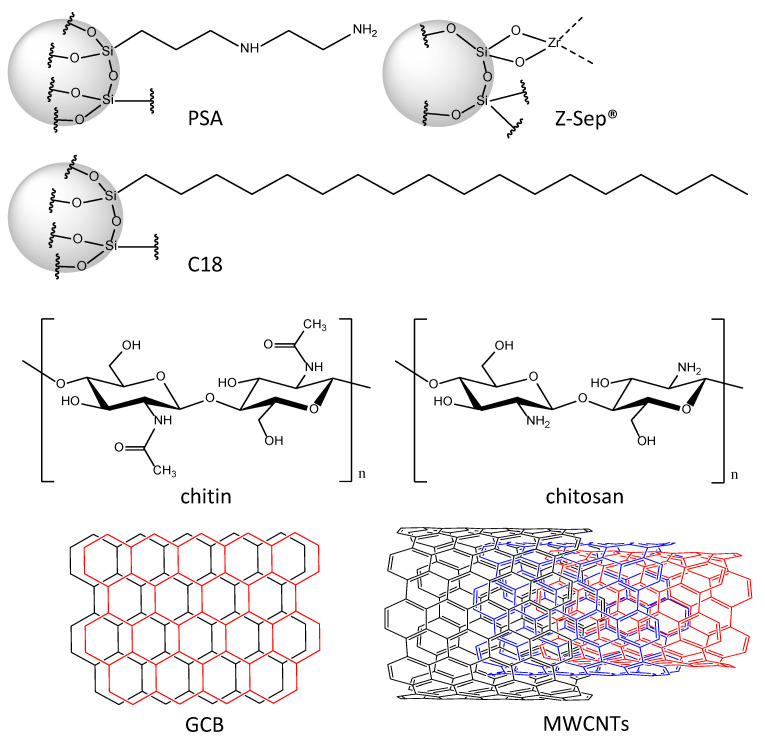
Commonly used dSPE sorbents.

**Figure 2 molecules-29-04656-f002:**
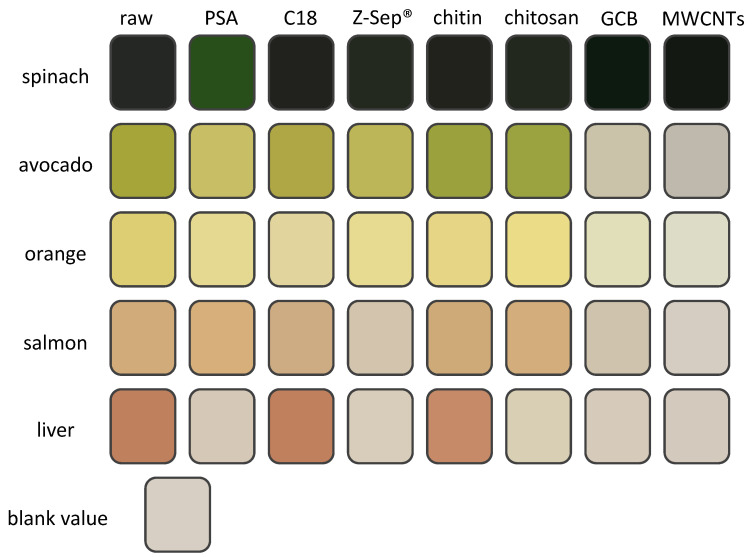
Color intensities from raw extracts compared to clean-up extracts after dSPE. The raw column shows the colors of the extracts before clean-up, and the other columns show the colors of the extracts after clean-up with the respective dSPE sorbents. A reduction in color intensity or change in color tone represents the removal of matrix components (especially of pigments). Colors were extracted from the pictures in Appendix A, shaped, and rearranged for better comparability.

**Figure 3 molecules-29-04656-f003:**
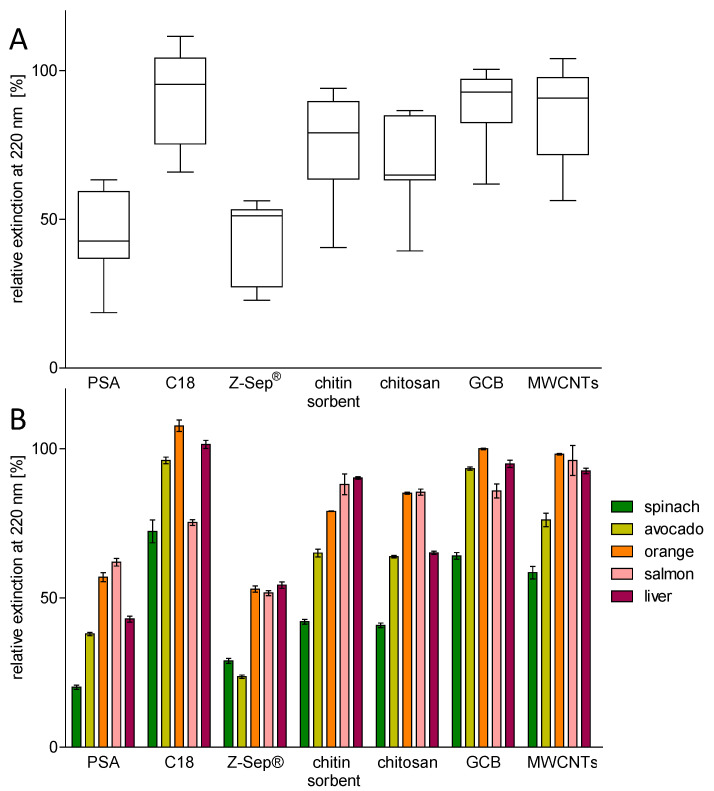
Distribution of the extinction of cleaned-up extracts at 220 nm relative to the extinction of the raw extracts, sorted by the sorbent used for clean-up across all matrices ((**A**), median indicated by the line in the box, whiskers show maximum and minimum values) and for the individual matrices ((**B**), n = 3, error bars show the relative standard deviation).

**Figure 4 molecules-29-04656-f004:**
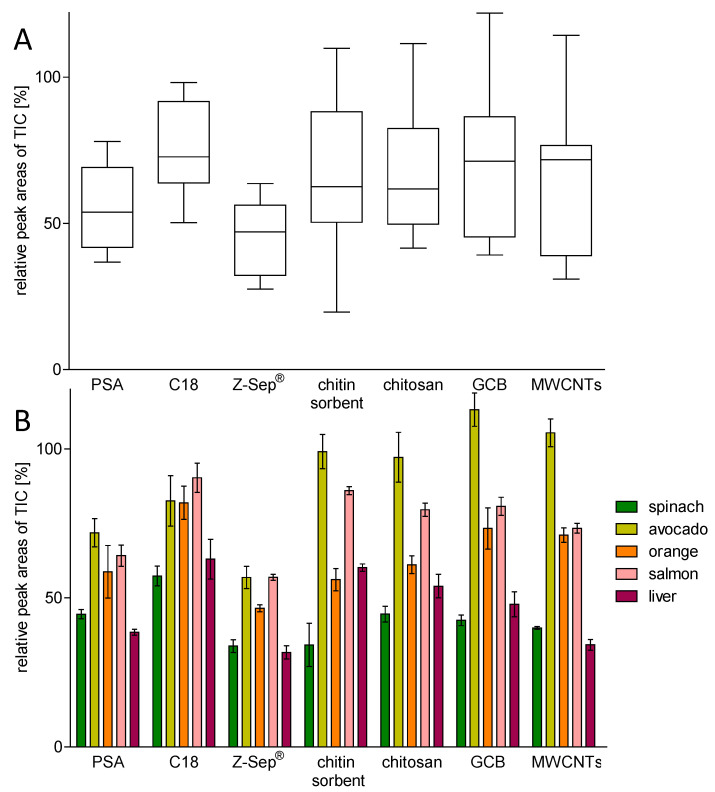
Distribution of mean peak areas in TIC chromatograms (scan *m/z* 100–600) of cleaned-up extracts relative to raw extracts, sorted by the sorbent used for clean-up across all matrices ((**A**), median indicated by the line in the box, whiskers show maximum and minimum values) and for the individual matrices ((**B**), n = 3, error bars show the relative standard deviation).

**Figure 5 molecules-29-04656-f005:**
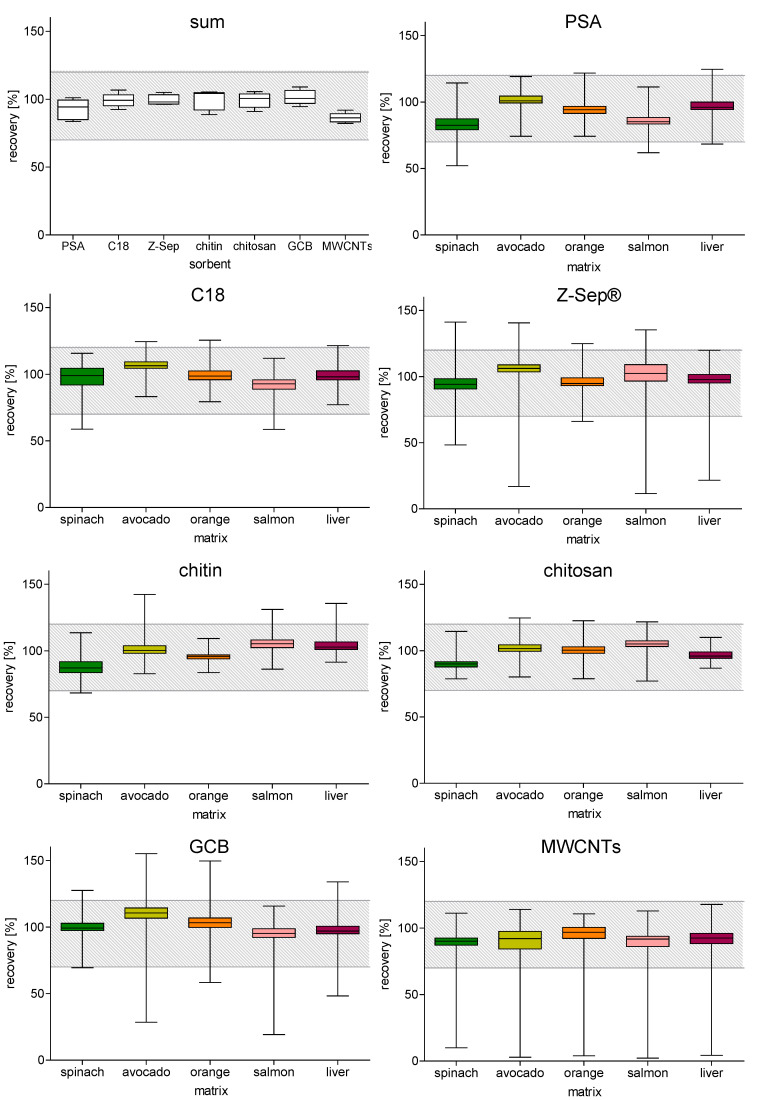
Analysis of the recovery data. For each matrix cleaned-up with each sorbent, the mean recovery across all analytes was calculated, resulting in one value per matrix and sorbent. The box plots represent the recoveries for the five different matrices, the lines represent the median recoveries, and the whiskers show the minimum and maximum values. The table containing all individual recovery values can be found in Appendix A.

**Figure 6 molecules-29-04656-f006:**
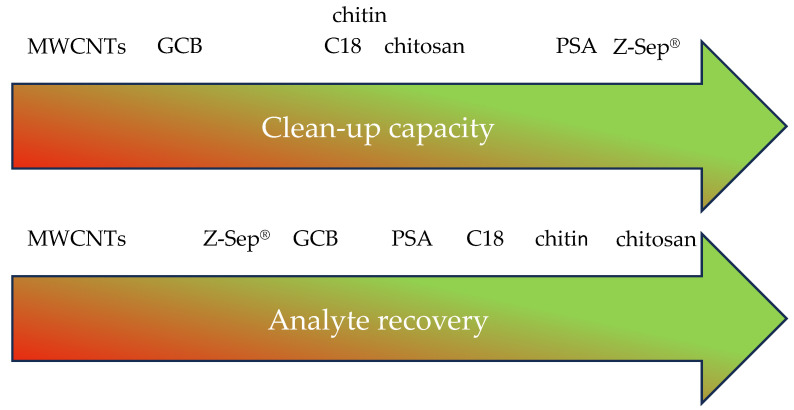
Rankings of seven tested sorbents regarding their performance in clean-up capacity experiments and affecting analyte recovery.

**Table 2 molecules-29-04656-t002:** Compositions of dSPE sorbent mixtures.

Sorbent	Amount of Sorbent [mg]	Amount of MgSO_4_ [mg]
PSA	50	150
C18	50	150
Z-Sep^®^	50	150
chitin	50	150
chitosan	50	150
GCB	7.5	150
MWCNTs	7.5	150

**Table 3 molecules-29-04656-t003:** Ratios between added water [mL] per g sample matrix in relation to the sample matrix.

Matrix	Homogenized Matrix [g]	Added Volume H_2_O [mL]
spinach	10.00	0
orange	10.00	0
avocado	5.00	6
liver	5.00	5
salmon	5.00	5

## Data Availability

Data are available from the corresponding author.

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
