# Peer review of "Systematic Comparison of Extract Clean-Up with Currently Used Sorbents for Dispersive Solid-Phase Extraction"

_molecules, 2024, doi:10.3390/molecules29194656_

Round 1

Reviewer 1 Report

Comments and Suggestions for Authors This manuscript primarily explores the purification effects of various adsorbents, including PSA, C18, Z-Sep®, chitosan, GCB, and MWCNTs, on different sample matrices such as spinach, avocado, orange, salmon, and liver during dSPE treatment. The study compares the color intensity, relative extinction at 220 nm, and total ion current (TIC) peak area of samples after purification with these adsorbents. Based on the experimental results, the manuscript evaluates the effectiveness of each adsorbent in purifying different sample matrices. The content of the manuscript is very rich and provides important assistance for sample pretreatment and food analysis fields. Although the content of this manuscript can provide great assistance for the selection of dSPE extractants, there are still some minor issues in the manuscript, and I believe that this work can be accepted with these modifications. Further comments: 1. The manuscript needs to be carefully checked to correct some formatting errors, such as removing annotations on page 9. 2. The author needs to provide GC-MS analysis spectra in the manuscript (or supporting materials). 3. How to achieve color transition between Figure 2 and Figure S2? Is this transformation accurate? 4. as shown in abstract that the Z -Sep was the best dSPE sorbent regarding clean-up capacity.   5. As mentioned in the abstract, the authors consider Z-Sep® to be the best dSPE sorbent in terms of clean-up capacity. What is the reason for this?

Reviewer 2 Report

Comments and Suggestions for Authors

This study has compared a few sorbents with different matrices. The idea is good but there are few my observations which I described in below points.

Abstract: The abstract is written very general. It needs to be specific and all the information is qualitative. There should be some data to be shown in the abstract. Additionally, there is a lack of information on the analytes used in this study. Also, keywords need to be redefined.

Introduction: This section needs improvisation by including more information about different types of analytes with the efficiency of sorbents with matrix intervention. Also, it is required to include the literature regarding other techniques and how they are inferior to QuECHErs and related to this study.
Result and Discussion: The figures are not self-explained. fig 2 needs to be defined more logically. Many results are not included in the discussion part. Additionally, 

Materials and Methods: Lack of scientific significance. 

Conclusion: It is too descriptive and needs to be paraphrased by reducing the word size and it should be concrete.

Reviewer 3 Report

Comments and Suggestions for Authors

Review manuscript ID: molecules-3203276

“Systematic comparison of extract clean-up with current used sorbents for dispersive solid phase extraction”

Authors describe the comparison of the advantage and disadvantage of use of different sorbent for solid phase extraction. The manuscript is well written, purification and recovery experiments are clearly explained and justified. I recommended the publication of this article with minor correction that are reviewer suggestions to further underline the importance of their work. English is perfectly understandable

Manuscript

Line 30, 53, 56,349,351,371,411,412, 520: DIN EN 15662. The wording DIN is correct but used only in the German version. To make the citation international it is recommended to write only "EN 15662"

Line 35, 536: the citation number 8 is incorrect. The article reported in bibliography is about determination of pesticide residues in fish tissues, not chicken.

Line 37: the word “concept” has been used several times in the first two paragraphs. It is recommended to utilize a synonym. For example in this line the authors can place "method".

Line 39: The citation should be reported only in line 41.

Line 56: the authors should use a synonym for “DIN EN 15662” such as “the regulatory method”

Line 66: Up to this point, the Authors describe the sorbent used in EN 15662. It is recommended insert a short phrase to introduce the next type of sorbent and specify that these are not used in EN method.

Line 101: The authors should use a synonym for “method”.

Line 109: Although the authors have specified that they have chosen the matrices to cover a broad range of applications of the QuEChERS approach, but considering that the method described in EN 15662 is focused on the analysis of plant-based foods and cereals, why were no matrices belonging to the cereals class included in the study? Maybe the authors can take into account of expand the study to these matrices.

Line 129: This part has already been said on line 104. The authors should include this paragraph above, writing only that validated and published methods were used (keeping the citations) for the matrices and molecules considered in this study.

Line 157: The phrase is already written in line 148.

Line 179: Why is C18 not named despite having poor results like those of GCB and MWCNTs (shown in figure 3)?

Line 188: Justified text of paragraph 2.1.3

Line 216: Why is C18 not named?

Line 374: The paragraph number is 3.2.4

Line 393: It would have been interesting to observe the effect of different sorbents in a wider range of wavelengths. The choice of UV-VIS analysis only at 220 nm was made because most components absorb at this wavelength or also for instrumental limits?

Line 409: It is not clear whether the same extracts were used for all tests (UV-Vis GC-MS/MS and recovery). If yes, add the spiking step in this paragraph and specify the concentration at which the analytes have been added. (See comment line 447)

Line 423: Replace 3.2.3 with 3.2.4

Line 447: Although the tests do not concern the sensitivity of the method, but only the recovery factor, one interesting fact is the concentration at which the extracts were prepared after purification and the dilution factor of the extraction method used.

Line 260: From the data in table S2, avobenzone has a recovery of 67% for the salmon matrix and not for the spinach matrix. (See comment line 264-277)

Line from 264- to 277: This concept is not clear. For the molecule avobenzone a recovery of 67% is shown, for the matrix spinach (actually salmon). The authors attribute this value to a potential nucleophilic addiction reaction catalyzed by acid components. This argument is not supported by the recovery value in the extract of the orange matrix (the most acidic among those analyzed) which shows a recovery of 90% (in table S2 89%), which cannot be considered as a low recovery value. Given that 67% is slightly under the criterium, we recommend reconsidering this hypothesis.

Line 287: 69% in orange not corresponded with value in table S2 (93%), avocado show this value.

Line 288: 63% in orange not corresponded with value in table S2 (83%), avocado show this value.
